# Nurses’ Perceptions of Ethical Conflicts When Caring for Patients with COVID-19

**DOI:** 10.3390/ijerph20064763

**Published:** 2023-03-08

**Authors:** Pedro Ángel Caro-Alonso, Beatriz Rodríguez-Martín, Julián Rodríguez-Almagro, Carlos Chimpén-López, Cristina Romero-Blanco, Ignacio Casado Naranjo, Antonio Hernández-Martínez, Fidel López-Espuela

**Affiliations:** 1Health Service of Castilla-La Mancha, Integrated Care Management of Talavera de la Reina (Toledo), 45600 Talavera de la Reina, Spain; 2Department of Nursing, Physiotherapy and Occupational Therapy, Faculty of Health Sciences, University of Castilla-La Mancha, Avd/Real Fábrica de Sedas s/n, 45660 Talavera de la Reina, Spain; 3Department of Nursing, Physiotherapy and Occupational Therapy, Faculty of Nursing, University of Castilla-La Mancha, Avd/ Camilo José Cela s/n, 13071 Cuidad Real, Spain; 4Department of Medical-Surgical Therapy, Psychiatry Area, Faculty of Nursing and Occupational Therapy, University of Extremadura, 10003 Caceres, Spain; 5Neurology Department, Universtity Hospital of Cáceres, 10003 Caceres, Spain; 6Metabolic Bone Diseases Research Group, Department of Nursing, Faculty of Nursing and Occupational Therapy, University of Extremadura, 10003 Cáceres, Spain

**Keywords:** clinical ethics, COVID-19, duty of care, nursing ethic, SARS-CoV-2, qualitative research

## Abstract

The COVID-19 pandemic has caused ethical challenges and dilemmas in care decisions colliding with nurses’ ethical values. This study sought to understand the perceptions and ethical conflicts faced by nurses working on the frontline during the first and second waves of the COVID-19 pandemic and the main coping strategies. A qualitative phenomenological study was carried out following Giorgi’s descriptive phenomenological approach. Data were collected through semi-structured interviews until data saturation. The theoretical sample included 14 nurses from inpatient and intensive care units during the first and second waves of the pandemic. An interview script was used to guide the interviews. Data were analyzed following Giorgi’s phenomenological method using Atlas-Ti software. Two themes were identified: (1) ethical conflicts on a personal and professional level; and (2) coping strategies (active and autonomous learning, peer support and teamwork, catharsis, focusing on care, accepting the pandemic as just another work situation, forgetting the bad situations, valuing the positive reinforcement, and humanizing the situation). The strong professional commitment, teamwork, humanization of care, and continuous education have helped nurses to deal with ethical conflicts. It is necessary to address ethical conflicts and provide psychological and emotional support for nurses who have experienced personal and professional ethical conflicts during COVID-19.

## 1. Introduction

The COVID-19 pandemic has had a dramatic personal and professional impact on health professionals, confronting them with multiple ethical challenges and dilemmas. This involved facing decisions in which important ethical values, principles, or duties conflicted or they had to make mutually inconsistent choices or actions [1]. At times, this involved choosing the “least bad” option [2,3].

At the beginning of the pandemic, ethical conflict and decision-making were influenced by the lack of materials (protective personal equipment (PPE) and respirators) and a shortage of resources—primarily beds and critical care professionals with intensive care unit (ICU) training [4].

Nurses who have been on the “front line” have perceived morally distressing situations or ethical dilemmas, such as the inability to provide adequate care, the provision of care in unsafe conditions, the lack of access to health care and to health care services [5,6], scarce resources, a high risk of infection, and concern about infecting their families; inadequate working conditions and rapid changes in workflow; a lack of specific skills and competences, fear of uncertainty, and frustration; and social stigmatization, isolation, and a lack of social support [6,7,8].

In this context, nurses encountered new tensions related to traditional ethical principles: social justice versus individual patient autonomy or beneficence versus public health non-maleficence [9]. The inability to involve patients and their families in decisions about their process and to accompany their patients in their suffering was a major source of ethical conflict [4]. Furthermore, the ineffectiveness of initial treatments, the “try anything and everything” strategy, and the use of age as a criterion for not admitting patients to the ICU (i.e., prioritizing resources for patients with a higher survival rate) also caused moral distress [4].

In addition, the situations experienced clashed with the core values and virtues attributed to nurses and with what is set out in their codes of ethics (humanity, compassion, benevolence, personal and professional competence, ethical practice, and moral rectitude) [10].

It is known that the psychological and emotional responses associated with discomfort or moral distress from ethically wrong actions can increase sometime after the situation [11,12]. These responses include feelings of helplessness, self-blame, anger, frustration, exhaustion, anxiety, and depression [13,14,15]; deterioration in the quality of care and teamwork; and the desire to quit the job [16]. The COVID-19 pandemic has been one of the most significant challenges faced by health workers, provoking intense negative emotional responses [16].

This study aimed to understand the perceptions and ethical conflicts faced by nurses working on the frontline during the first and second waves of the COVID-19 pandemic and the main coping strategies they used to resolve these conflicts.

## 2. Methods

### 2.1. Design and Participants

A qualitative study was designed and analyzed using Giorgi’s descriptive phenomenological approach, which discovers the meaning of a phenomenon through the identification of fundamental themes [17]. This approach was chosen to describe the experiences of nurses caring for people with COVID-19 during the first two waves of the COVID-19 pandemic and the ethical dilemmas from the nurses’ experience through a phenomenological analysis of (and in) their own words [18]. This study followed the recommendations of the Consolidated Criteria for Reporting Qualitative Research (COREQ) guidelines [19].

This study used a theoretical sample of nurses who cared for people admitted with COVID-19 during the first and second waves of the pandemic in public hospitals of the regions of Madrid and Extremadura. These two regions were chosen because they were among the most affected during the first two waves.

The inclusion criteria were as follows: (1) nurses who had worked for more than two months in active duty during these two waves of the pandemic caring for patients suffering from COVID-19; (2) nurses working in hospital units or intensive care units. The exclusion criterion was as follows: (1) nurses who had worked for less than two months with COVID-19 patients. The socio-demographic characteristics and characteristics related to educational level and employment of the participants are shown in Table 1.

Semi-structured interviews conducted between May and December 2020 were used for data collection to include nurses’ experiences of the first two waves of the COVID-19 pandemic. Interviews lasted an average of 66 min (range 36–116 min). Semi-structured interviews were chosen for their ability to provide a “naïve description” of this psychological phenomenon in the participants’ own words [18,20].

### 2.2. Data Collection and Data Sources

Data collection and analysis were carried out simultaneously following the constant comparison method. Data collection continued until data saturation, at which point additional input (new codes, categories, or subcategories) from new participants no longer provides new information [21].

The interviews were conducted in a comfortable, private location agreed upon with the participant by the main researcher (blinded for review), who had a script of topics that could appear openly during the interview (Table 2) [20]. The interview script of topics was refined throughout the study, following the constant comparison method, with the information that emerged in each interview. The interviewer was an expert in qualitative research and had extensive experience in the semi-structured interview technique. He has a background in Public Health and no clinical experience in the study field. The position of the interviewer is a strength of this study, which prevented the interviewer from knowing too much about the phenomenon under study. Moreover, none of the authors knew the participants. Most of the interviews were conducted face-to-face; however, due to the evolution of the pandemic, four participants preferred to conduct the interview via videoconference.

### 2.3. Data Analysis

Once transcribed and anonymized, the interviews were analyzed following Giorgi’s phenomenological method: (1) collecting and describing phenomenological data, (2) reading the entire description, (3) breaking descriptions into meaning units, (4) transforming meaning units, (5) identifying the essential structure of the phenomenon, and (6) integrating features into the essential structure of the phenomenon. Theoretical reflective writing was used during the analysis process [17].

The data analysis was carried out independently by two researchers, who subsequently reached a consensus on the results; in the event of a disagreement, a third researcher was used. The research team examined the codes and themes to formulate the categories and final themes. Atlas-ti 8.0 software was used to assist in this phase.

### 2.4. Trustworthiness

The criteria of credibility, transferability, dependability, and confirmability were used to establish study trustworthiness [22]. Various strategies have been developed to ensure credibility, such as the following: reflexivity (all researchers had a constant reflective attitude attending to methodological crossroads decision-making and interpretative issues), research triangulation (comparison and discussion between two different researchers during the analytical process; in the case of discrepancy, a third investigator was called upon to resolve the discrepancies), and return to interviewees (during the last interviews, as themes were becoming saturated, new questions were introduced in interviews to confirm the intersubjective experience interpretation; in addition, the transcripts of the interviews were returned to the participants to check their agreement). In relation with confirmability, along this manuscript, we have tried to be very transparent with our research process for other researchers to know what stages and how we have to cope with different tasks and difficulties. Finally, in terms of helping the audience to assess the transferability of results, we have meticulously described the participant characteristics that condition their experience, as well as the environment where the research was conducted.

## 3. Results

Table 1 shows the main characteristics of the 14 nurse participants. During the first and second waves of the COVID-19 pandemic in Spain, the nurses working in hospital units or intensive care units faced significant ethical conflicts, identifying two main themes: (1) ethical conflicts experienced during the pandemic on a personal and professional level and (2) coping strategies used to deal with ethical conflicts (active and autonomous learning, peer support and teamwork, moments of emotional catharsis, focusing on care, accepting the pandemic as just another work situation, forgetting bad situations experienced, valuing positive reinforcement from patients, relatives, and society, and humanizing the situation). Table 3 and Table 4 show the categories, codes, and selection of the main verbalizations for each of the emerging themes. The verbalizations are accompanied by the code of each interview, which identifies the interview number, the gender (F: female and M: male), and age of the participant.

### 3.1. Theme 1. Ethical Conflicts during the Pandemic

The nurses faced ethically demanding situations and ethical conflicts arising from a lack of knowledge, information, action protocols, staff, or protective materials. These ethical conflicts arose in both their personal and professional lives (Table 3).

### 3.2. Ethical Conflicts at the Personal Level

Faced with the fear of infecting their family members and the fear of the contagion, most nurses made significant changes to their personal lives, including modifying their family routines and personal behaviors, voluntarily isolating themselves from family and friends, and living with other health professionals as protective measures. In addition, at the onset of the pandemic, nurses sacrificed their personal lives in order to practice their profession full time, which sometimes led to family conflicts when their relatives asked them to leave their jobs.

### 3.3. Ethical Conflicts at the Professional Level

The ethical conflicts that stood out at the beginning were those related to moral suffering, the perception of risking personal integrity and life, the wrongful death of patients, and the neglect of patients’ basic rights.

#### 3.3.1. Moral Suffering

The nurses experienced moral distress due to external factors that prevented them from providing the care they considered ethically correct.

#### 3.3.2. Endangering Their Personal Integrity and Their Lives

The nurses continued to provide care, despite the absence of protective equipment, working with non-approved equipment or not knowing what they were dealing with, which could put their right to personal integrity and their own lives at risk for the benefit of the general population, thus exceeding the obligations of a nurse according to the legislation in force and the code of ethics.

In addition, they highlighted the conflicts that arose with their superiors over certain orders that put their health at risk, for example, the recommendation not to wear masks during the early days of the pandemic so as to avoid being disconcerting patients.

#### 3.3.3. Poor Patient Death

One of the aspects that most marked the nurses at the beginning of the pandemic was the “bad death” or “poor death” experiences of the patients, due to the rapid decompensation they suffered and dying in solitude. The solitude faced by patients occurred not only because family visits were prohibited, but also because the work saturation meant that when they returned from making the rounds of the rooms, some patients had died, often showing signs that they had done so in distressful situations (this included facial gestures, patients who had fallen on the floor after getting out of bed, etc.).

#### 3.3.4. Neglect of Basic Patient Rights

The nurses highlighted the neglect of basic patient rights related to autonomy, beneficence, and justice.

##### Autonomy

On certain occasions, nurses did not respect the principle of patient autonomy in order to protect community health, for example, by using physical restraints on people with intellectual disabilities to prevent them from leaving the room, in this case taking precedence over the community good.

##### Beneficence

Especially at the beginning of the pandemic, despite the effort to do everything possible with the knowledge they had available, nurses noted the rapid deterioration of patients and the high death rates. This provoked intense feelings of frustration and helplessness in the nurses. On other occasions, the nurses acknowledged a sense of guilt and questioned whether they could have done more to save the patient.

In these initial phases, the nurses’ questioned the care provided due to a lack of knowledge of the virus, lack of effective protocols, fear of the contagion, and lack of materials. They considered that the care provided was not the most appropriate or did not provide all the support and comfort to the patients.

Moreover, especially in the early stages, nurses had to care for patients with COVID-19 without adequate protective measures to protect themselves or other patients from infection.

The nurses acknowledged that on certain occasions, the information provided to relatives was concealed or embellished on the recommendation of their superiors, believing that this would save them suffering; although this lack of honesty made them feel like impostors.

##### Justice

In the initial phases, the principle of justice was affected by limited resources and by the need to prioritize care for COVID-19 patients and treatments. Furthermore, in many cases, the protocols were unclear. The nurses emphasized that limited resources and the large number of patients requiring such resources made the criteria for admission to ICU beds very restrictive. Among the criteria for prioritization, in the early phases of the pandemic, age was used, with young patients being prioritized over older patients. Moreover, in the early phases of the pandemic, patients with COVID-19 were considered non-reuscitable, and therefore sometimes died of other diseases, such as cardiac pathologies, without nurses being able to attempt cardiac resuscitation. Moreover, nurses noted that routine care was often delayed or curtailed to redistribute scarce resources at the onset of the pandemic.

In addition, the nurses themselves perceived that they had suffered from this inequity in access to protective equipment between different hospital units and even between different professional categories within the same unit. Thus, nurses reported that physicians had access to better protective equipment than themselves, even though nurses cared for patients at their bedside 24 h a day.

Also noteworthy was the perceived lack of solidarity among the nurses themselves, since they did not share the material and kept it locked away to avoid others using it, when previously it had always been shared among different units and professionals.

In addition, they perceived an inequal exposure to the virus among doctors and nurses, since nurses spent more time in an infectious environment, indeed, they cared for patients 24 h a day, and the care techniques they performed were more invasive and riskier. In the early stages of the pandemic, the nurses considered that physicians delegated upon nurses’ actions outside their scope of competence, such as the responsibility of assessing patients to prevent them from entering rooms and taking risks. In addition, nurses perceived that their professional relationships with physicians changed during the pandemic, with contact, visiting times, and meetings decreasing due to physicians’ fear of the contagion. In addition, according to the nurses, physicians in some specialties were allowed to telework.

Finally, the nurses felt that the healthcare system had failed because of the lack of resources for themselves and patients to be treated with equity and fairness.

### 3.4. Theme 2. Coping Styles for Dealing with Ethical Conflicts

Faced with the above ethical conflicts, the nurses used the following coping strategies.

#### 3.4.1. Autonomous Active Learning or Learning among Peers

Due to the lack of information and training in the institutions where they worked, the nurses used active autonomous learning and peer learning. In addition, the lack of institutional protocols, especially at the beginning of the pandemic, led them to develop their own protocols, which they modified as the situation changed.

As self-training resources, they used social networks, such as YouTube, Twitter, WhatsApp groups, or internet resources, specifically to learn novel techniques or in cases where they had to urgently start working in a new unit. This self-training was conducted by the nurses in their free time or outside of their workday.

#### 3.4.2. Peer Support and Teamwork

To deal with ethical conflicts, nurses considered peer emotional support and teamwork essential, including nurses and nursing care technicians within the nursing team.

Regarding teamwork, the nurses highlighted the trust they had in the team members, perceiving it to be greater than they had ever experienced and how this support could continue outside the working day through telephone conversations or WhatsApp groups, which were conceptualized as an outlet to share the experiences and problems they were experiencing, since on many occasions, they did not want to worry their own family.

#### 3.4.3. Moments of Catharsis and Disconnection through Humor, Music, or Dance

As an escape from the difficult situations they endured, the nurses used a sense of humor, music, or choreographed dances with other colleagues at specific moments of their working day. The nurses emphasized that these were only occasional moments of the working day and that they had nothing to do with the frivolous image of dances that the media have conveyed in Spain, which is far removed from the reality experienced by the nurses during the pandemic.

#### 3.4.4. Staying Focused

At the beginning of the pandemic, the work environment did not help the nurses due to chaos, disorganization, and the lack of information, protocols, and protective equipment. Moreover, during this initial chaos, the nurses felt a lack of support from their superiors (nursing supervisors, hospital management, health administration, etc.), which meant that they had to resolve the conflicts that arose as best they could. Faced with this situation, the nurses focused on care and on the problems that arose at any given moment as coping strategies.

#### 3.4.5. Accepting the Pandemic as a Work Situation

Nurses perceived the pandemic as a work situation that they had to endure, something historic and which they had to face as their duty and professional commitment.

#### 3.4.6. Forgetting the Bad Situations That Had Occurred

A protective mechanism in situations of post-traumatic stress was dissociative amnesia (forgetting moments with a high negative emotional charge), trying to forget the bad moments experienced.

#### 3.4.7. Valuing the Positive Reinforcement of Patients, Their Families, and Society

The gratitude and positive reinforcement received from patients, their families, and society, helped them cope with the difficulties and gave them encouragement during the first wave of the pandemic, perceiving that society valued their efforts and the risk they were exposing themselves to in order to care for others.

A notable occurrence in Spain during the confinement was that every day at eight o’clock in the evening, people spontaneously went out to their windows to applaud health professionals and other first responders. According to the nurses’ speeches, this applause helped them to cope with the tough working situations they were experiencing, although at the same time, they demanded that this support be translated into real improvements in their working conditions and in healthcare at large.

#### 3.4.8. Humanizing the Situation

Faced with the harshness of the situation and the violation of patients’ rights on multiple occasions, the nurses tried to humanize care as much as possible. They spent time during their working days, and often outside of them, making phone and video calls with relatives. In addition, as more information was known about the virus, the nurses tried to allow the families to say goodbye to their family member by allowing them quick visits for which they provided them with PPE that they did not have to spare at the time.

## 4. Discussion

This study analyzes the ethical and moral concerns and ethical conflicts experienced by nurses during the COVID-19 pandemic. Two main themes emerged: the ethical conflicts experienced on a personal and professional level; and the main coping strategies used by nurses to deal with these conflicts: learning; support and teamwork; catharsis and disengagement; focusing on care and everyday work; acceptance of the pandemic; “forgetting”; social positive reinforcement; and humanizing care.

The setting in which the study took place is a COVID-19 care unit (ward or ICU), which is similar to the setting of many other studies worldwide [23]. In our case, data saturation was reached with 14 nurses, but other studies present a lower sample size (seven nurses) [23].

### 4.1. Theme 1: Ethical Conflicts during the Pandemic

In line with previous studies, the first-line nurses decided, voluntarily and in agreement with their relatives, to adopt measures to protect them from the possible contagion, such as isolating themselves from their relatives at home, living in other places or with colleagues, or modifying family routines (shower on arrival at home, extra washing of clothes, etc.) [16,24,25,26]. This is a source of family conflict and increases the nurses’ emotional burden and feelings of isolation and loneliness [8,27,28].

In addition, the pandemic situation, uncertainty, fear, and lack of protection in some instances led nurses to consider a temporary career break or request sick leave [25,26]. However, nurses showed moral courage, understood as the determination to stand firm to do what is ethical when doing the right thing is not easy, an ethical virtue that helps someone overcome physical or emotional obstacles to act on what is believed to be the right thing [18,29,30,31]. In this manner, most nurses decided to fulfill their duty and obligation of care to patients with COVID-19 despite fear [16,30,31]. Other reasons given that follow the line of previous studies include the deep-rooted professional commitment, nurses’ strong vocation, their personal values, or their commitment to society [25,26,32].

Our participants accepted the pandemic as just another work situation they had to face, coping with and assuming responsibility for the care of people with COVID-19. Nurses have reported feelings of pride, considering that they are better professionals after having experienced this situation [16]. This result can be explained by the fact that, despite the changing clinical context during the first waves, they were able to adapt quickly to the changes and to the new way of working [25,31].

As other studies have already shown, the moral suffering of nurses during this pandemic is remarkable and increases as the pandemic progresses, with loneliness and the isolation of patients and families acting as aggravating factors, or the prioritization of resources due to the scarcity of ventilatory support, personal protective equipment (PPE), ICU beds, etc., which put the personal integrity of the nurses at risk [4,33,34].

Another highly traumatic fact was the elevated number of deaths and the circumstances in which they occurred, conceptualized by the nurses as “poor deaths”, not only because patients had to die alone, without family members and without traditional end-of-life practices, but also because of how these deaths sometimes occurred (shortage of ventilators, cardio pulmonary resuscitation was not performed, etc.), causing extra moral suffering to the nurses [16,28,31,35].

Moreover, nurses considered that despite doing everything possible to save the patient’s life or to provide excellent quality and ethical care, the pandemic situation prevented them from providing care as they would have liked [8,9,36]; this generates feelings of frustration and helplessness, considering that the situation became unsustainable [16,35]. In addition, they sometimes felt guilty about the care provided, which can be detrimental to their mental health and cause added moral damage [4,8,31].

Nurses highlighted the violation of the fundamental ethical principle of justice as the pandemic forced the values of autonomy and beneficence to yield to justice [9]. The shortage of material resources (ventilators, ICU beds, etc.,) is considered by nurses to be a major ethical burden, having to face situations in which they knew the ethically correct actions, but were unable to perform them. This led to decisions that generated moral or ethical discomfort, due to inequity in the access to resources and treatments (respirators, ICU, or Intermediate Respiratory Care Unit (IRCU) beds, etc.) [25,35,37,38,39]. The pandemic led to a significant increase in the volume and intensity of work, which undermined equity in access to resources and treatment [9,25,40]. Consequently, complaints from healthcare professionals about the lack of protective gear have been universal [25].

Thus, the nurses reported having to buy their own equipment at times or having to behave in an unsupportive manner towards colleagues from other units by not lending them protective materials [9]. Faced with this situation, nurses have highlighted the insufficient institutional support or negligence in protection of staff [25,35].

Other studies highlight the inequality in the exposure to the virus, showing that nurses have been more exposed to the virus than any other healthcare profession [9,15,27]. The pandemic situation caused them to assume the functions of other professionals (physicians, technicians, etc.), increasing their responsibility and feelings of isolation [16,31]. In addition, this was perceived as a complete lack of respect and consideration for the nurse as an expendable person [16,27].

In addition, the fact that family members could not accompany the patient during admission meant that nurses facilitated communication between patients and family members through video calls, which meant an overload of work and an emotional and moral burden since they became witnesses to conversations and farewells that were previously done behind closed doors [9,16,30,41]. This situation provoked another novel ethical conflict, such as the communication of death via telephone to the relative, and how, applying the principle of beneficence, the nurses tried to relieve suffering of the relatives and therefore softened the information provided about the patient’s death so that they would have a better memory of the time or less grief, and this “white lie”, which was said out of compassion, made them have second thoughts, although it was relieving them to think that they were doing so to avoid suffering.

### 4.2. Theme 2: Coping Styles for Dealing with Ethical Conflicts

The COVID-19 pandemic was characterized by real-time experience and rapid and constant changes, including contradictory messages from government agencies that exacerbated fear and mistrust and difficulties keeping up with the pandemic [31]. Nurses were concerned about their training on an ongoing basis and, especially outside working hours, attempted to update their training through social networks and protocols sent by hospital managements [7,16,25]. This continuous learning has led to changes in the organization of care to make it more efficient and safe [18,33]; this shows that, as nurses became more familiar with the care of the disease, the emotional burden and fear decreased [25,32].

The nurses considered it very positive to feel the emotional support and trust between colleagues, and they emphasized that teamwork is vital to survive this situation and avoid negative emotions, where this support continued outside the working day [16,27,31,42].

In addition, many nurses faced the situation with a positive attitude, allowing them to reinterpret the negative situations experienced [43,44]. They emphasized that they did not regret having chosen the profession, accepting this situation as part of being a nurse and the duty of care [25]. Another strategy to reduce anxiety levels was the use of humor [45,46], in what they called catharsis, although they complained about how the media or the public sometimes misinterpreted these venting behaviors.

Disengagement and venting represent dysfunctional or unhealthy coping behaviors, such as denial, avoidance, or giving up [47], and are used in stressful situations [48]. These strategies of emotional disengagement and venting behaviors have been observed in other studies to ease mental discomfort [49].

However, another study describes unhealthy coping, such as substance abuse, suppression of their feelings, “burying the trauma” (forgetting this bad experience), and emotional detachment [16].

Another key aspect was to feel the support of patients, family members, and the community [16].

### 4.3. Limitations and Strengths

Among the strengths of this study, we analyzed the first and second waves of the pandemic, which were moments of great uncertainty and a lack of knowledge, in which ethical conflicts were more novel and prominent. In addition, the study included participants from various autonomous communities of Spain, based on different units and training, professional experience, and sociodemographic characteristics. The reviewed studies address ethical dilemmas; however, few studies refer to nurses’ coping strategies.

## 5. Conclusions

Nurses have experienced ethical conflicts on a personal and professional level and have strived to optimize and deliver safe and ethical care to patients with COVID-19 even when their lives were at risk. The main coping strategies used by nurses to deal with these conflicts have been active and autonomous learning, peer support and teamwork, moments of emotional catharsis, focusing on care, accepting the pandemic as just another work situation, forgetting bad situations experienced, valuing positive reinforcement from patients, relatives, and society, and humanizing the situation. Strong professional duty and commitment, teamwork, humanization of care, and continuing education have helped nurses cope with such conflicts. Ethical concerns and conflicts that arose along with work overload and other factors may have contributed to increased psychological distress among frontline nurses.

Future studies should analyze ethical conflicts arising in the context of primary care and social and health care. In addition, it would be useful to know which interventions would lead to improvements in the coping strategies used by nurses during ethical conflicts.

### Implications for Nursing Management

Ethical concerns and conflicts that arose along with work overload and other factors may have contributed to the increase in psychological distress among frontline nurses. It is necessary to address ethical conflicts and provide psychological and emotional support programs to those nurses who have experienced personal and professional ethical conflicts when trying to optimize and provide safe and ethical care during COVID-19.

## Figures and Tables

**Table 1 ijerph-20-04763-t001:** Main characteristics of participants (N = 14).

Participant Characteristics		*n*
Age	<30-years-old	2
	30–39-years-old	8
	40–49-years-old	2
	>50-years-old	2
Gender	Male	3
	Female	11
Highest academic qualification	Bachelor’s Degree	6
	Specialist	1
	Master	6
	PhD	1
Type of work	Temporary employment	9
	Fixed-term contract	3
	Permanent contract	2
Type of unit	Intensive care	6
	Emergency	2
	Medical unit	6
Change of unit during COVID-19 crisis	Yes	3
	No	11
Years in practice	0–4 years	2
	5–10 years	5
	11–15 years	3
	More than 25 years	4

**Table 2 ijerph-20-04763-t002:** Interview topic script.

Perceptions of nurses’ perceived ethical conflicts during pandemic care: major ethical dilemmas and conflicts.
Perceptions of nurses who worked on the front line caring for people with COVID-19 on the coping strategies they used to deal with ethical conflicts.
Perceptions of the impact of ethical conflicts on the personal lives of nurses who worked on the front line caring for people with COVID-19.
Nurses’ perceptions of the impact of ethical conflicts on the nursing profession in general and on their day-to-day lives as nurses.

**Table 3 ijerph-20-04763-t003:** Categories, codes, and verbalizations of Theme 1: Ethical conflicts experienced during the pandemic.

Categories	Codes		Verbatims	
Ethical conflicts at the personal level	Changes to familiar routines		“For the first fifteen days, for fear of infecting my partner, I voluntarily isolated myself in a room. I would come home from work, take two showers during the shift, during the break and at the end of the shift. When I got home, I went straight into the room without touching anything. My partner prepared a room for me, set up a TV, a small table for me to eat and I ate in the room. I did everything in the room. He would carry the tray of food to the door of the room and leave it on a chair for me. During this time, not a hug, not a kiss… these were forbidden.” (P. 07F32).	
	Voluntary isolation of family members		“I said <<my first measure will be to be apart from my family and to avoid them (her parents) from teaching any of it>>. I put on a mask, we ventilated the whole house, and they were getting used to a new way of living together… in the end I decided to rent an apartment, so I can be away from my parents so they could live.” (P. 02F45).	
	Living with other healthcare professionals		“All of us nurses from out of town lived with other nurses. I was living with fellow nurses; we were all in on it for COVID-19 as much as possible. We complied with the measures, you showered at work…you knew you weren’t going to hurt a family member who was out of your scope of work or anything, we were all isolated, we all understood each other. Some colleagues found it hard to rent an apartment or were even encouraged to move house when the pandemic broke out.” (P. 05M30).	
	Family conflicts		“My partner was crying and asked me not to go to work, not to leave the house. <<Don’t go out, don’t go to work>>.” (P. 07F32).	
	Temporary career break		“There are colleagues who have quit (taken a leave of absence) simply out of fear. It happened especially at the beginning. They had elderly parents or vulnerable people, what did they do with the parents? On the one hand, I understand it and on the other hand I don’t. Besides being a nurse, you can go to another place to live and have your husband take care of your parents, there are people who have left simply out of fear.” (P. 03F38).	
Ethical conflicts at the professional level	Moral suffering		“I don’t like to encounter dead people; I don’t like to see people suffering. I love my profession. I love helping people and being there. I’ve been there every day to take care of my patients, every day, there’s not a day that I haven’t gotten dressed, you know? I don’t know a single person who hasn’t stopped getting dressed for a day. I love grandparents calling your name and asking for your help. So, walking into the rooms and finding someone dead, it may have happened to you once, one night, I don’t know, but here it was, those rooms, those closed room doors, that you started down the hallway, when you got to the end and an hour and a half had passed, they could have been calling out to you and you didn’t hear it, you know, the old people because they were old, but the young people came with pneumonia in such a state that they were not able to ring a bell, they had no strength.” (P. 04F46).	
	Putting personal integrity and life at risk		“We began to lack equipment; we were already re-sterilizing the masks. There were days when we didn’t have any Filtering Face Pieces 3 (FFP3) masks, which was recommended for intubations, aspirations, etc… I remember the anesthesiologist telling us <<do you have a FFP3 on?>> and we responded <<no, because we don’t have any>>. When my partner and I came out of the shift we hugged each other and cried. We said <<we don’t know if we are going to die>>. Because the feeling we were working with was <<we are going to die>>.” (P. 07F32).	
	Poor patient death		“It took you three hours to do a round. If it was the first one you were doing, in the next round you found them dead and in a very bad way, they were all blue, or had fallen on the ground… and whoever tells you otherwise is lying. You couldn’t do anything. You went into the rooms and found the people dead … and in a very bad way and alone”. (P. 04F46).	
	Neglect of basic patient rights	Autonomy	“In people with intellectual disabilities we had to use five-point physical restraints. They were independent people, who moved around… it’s true that they had that disability. They were alone in a room twenty-four hours a day because neither family members nor caregivers came. And we couldn’t be with them inside because of the risk of infection. There was a risk that if you removed the restraints they would touch your face, remove the screen, cough… They were two boys with disabilities, as they were positive, you could not leave them “unrestrained” because we couldn’t risk them doing what they wanted to do at all times, such as leave the room. Therefore, we had to use restraints. For me, to see those kids with this hyperactivity and, giving them medication to make them more relaxed was very distressing. They had restraint marks on their hands from wanting to take them off. That was hard for me because they didn’t understand the situation…” (P. 05M30).	
	Abandonment of basic patient rights	Beneficence	Doing everything they can for the patient	“I’ve worked more hours than my fair share. I did everything I could. … We did the best we could and the best that we could do at the time. We tried everything, doses and medications were modified, and even so, we have also seen very young people die. But we have all been left with the feeling that we could not have done more…” (P. 06F52).
			Frustration and helplessness for not being able to save the patient	“I’ve cried a lot because of the helplessness of knowing that you weren’t taking care of them as you should because at the beginning there was so much fear that they were like patients who were hopeless. It didn’t matter how old they were, the first thing we were told was <<if someone stops (cardiorespiratory arrest) they are directly not resuscitable>>, and we knew that if you performed Cardio Pulmonary Resuscitation (CPR) on that person, because maybe they stopped for another reason, not for… maybe they would come out with CPR, but they were not resuscitable…” (P.07F32).
			Guilt	“What happened is that we couldn’t give them at the beginning the level of care that you would have given to another type of patient without COVID-19, because you thought it was so contagious, that everything was so dangerous. I am one of the nurses who shakes hands with the patients, even if I am in an isolation suit, with my gloves… I mean, touching their face, and combing their hair, we usually give them a very humanized treatment and suddenly we can’t even touch them, the farther away the better. I think that if there had been more information, the care would have been better, and it would not have generated so much frustration. You go home with a feeling of guilt for not having been able to take care of everything. I know that I have taken care of him, as best as possible, but it’s true that you go home feeling bad because there have been days when we have left at least 25 admissions pending because we had to go up to the ward and the entrance room was full because we just couldn’t take in any more people”. (P.03F38).
			Questioning care	“The first few days you’d get home and I, for example, who don’t consider myself a super nervous, super anxious person, I was very worried, did I do things right, did I take off my personal protective equipment (PPE) properly, did I take the right protective measures…?”. (P.01M24).
			Duty of care	“In the intensive care unit (ICU) we used the FFP3s per shift because it was safe. If there weren’t enough supplies, we bought them ourselves. If you use this type of mask you can’t afford to use the same one the next day because you are going to infect the other poor guy who just got out of the ICU because he is a little bit better or you are making the COVID-19 disease of whoever is suffering get longer over time because you are over infecting that person, or that person is over infecting you.” (P. 02F45).
			Sparing suffering to family members	“During the deaths, when the families called you and asked you <<but did you see that he was in pain, or that he suffered?>> sometimes you didn’t know what to say because of course… The supervisor told us <<you can’t tell the families that it was bad because that is the memory that will remain>> and I really had times when I said <<I have the feeling that I am telling people that no, that he was fine, and that the person was calm… when it isn’t true>>>. But they told us <<you can’t tell that to the families>>. Being on the other end of the phone, I understand it and you have the feeling of saying <<I am lying>>.” (P.03F38).
		Justice	Inequity of resources and treatment	“Maybe because you didn’t have time, some patients have died in their room, and you didn’t know if you were going to be able to take them out [to the ICU] or you couldn’t take them out because you had to prioritize. If there was a heart attack or respiratory failure, you had to leave them (they were not considered resuscitable), it happened a lot. And they died because of the infarction… There were no established criteria or protocols… But you more or less knew that you had to leave the patient on the ward because the ICU was collapsed. One criterion we used was age and previous pathologies.” (P.06F52).
			Inequity in the allocation of personal protective equipment	“We have realized that apparently some lives were worth more than others because the doctors, when there was no FFP3 for us, for the few times they came in, they came in with an FFP3. They were provided with better material, with more comfortable goggles, and we had ulcers and marks on our faces because we had goggles that were the same for everybody, right? it was our material and that was it. To me, those details hurt a lot because I didn’t expect that differentiation.” (P. 07F32).
			Lack of solidarity among professionals	“They came to ask for protective equipment from other units, and we told them that we wouldn’t give it to them because it was for us. We had never done it before, we nurses have always been supportive and this made us feel bad… but we couldn’t do anything else, on our floor they were all positive…” (P. 09F52).
			Inequality in exposure to the virus	“I think that the doctors have taken a step back, the nurses have been the ones who have moved forward, the ones who have managed everything. They have left us as the first ones to care, and then the doctor came. Those of us who entered the room were always the nurses and assistants. The doctors were very afraid, they did not enter the room… telling you everything from the doorway. There were things that they had to come into the room to do, but if they could, they avoided doing so. They trusted us, what we told them and what we didn’t, which we valued… It’s true that they gave us a lot of freedom, but of course the problem was that many times you were helpless. It was a totally self-interested trust and so they avoided going in to see the patient. To avoid contact with COVID-19, basically.” (P. 08M32).
			Failure of the health system	“Each death is not a failure on our part (nurses) or on behalf of the doctors, it’ s just that we couldn’t… I mean, we couldn’t do anything else. There were patients who had to go down to the ICU to be put on a respirator and if they didn’t go down you knew they were going to die. It’ s a failure of the health system, like everything else. It’s bad luck, because if it had happened fifteen days beforehand there would have been a ventilator for you and now there isn’t…” (P. 10F30).

**Table 4 ijerph-20-04763-t004:** Categories, codes, and verbalizations for Theme 2: Coping strategies for ethical conflicts experienced during the pandemic.

Categories	Codes	Verbatims
Active learning	Autonomous or peer learning	“We were very scared, we had no training as such, meaning that the Administration did not give us any course with training hours. All the training has been watching videos and documentation at home or learning with co-workers who accompanied you in putting on and taking off the personal protective equipment (PPE) and showed you the steps.” (P. 01M24).
	Use of social networks and internet resources	“I looked for information on the Word Health Organization (WHO) website, on social networks, especially on Twitter, I contacted other professionals from other intensive care units (ICUs) in Spain and they told me what they were doing, I looked for videos of other hospitals in Spain, or other hospitals internationally and I stuck with what I thought could be better or could be better adapted to our unit and that’ s how we were developing the protocols.” (P. 07F32) “We have a special WhatsApp group for COVID-19 where the supervisor passes on all the information, all the protocols that were in place.” (P. 09F52).
	Outside working hours	“My days off were to watch videos on the internet about respirators, to handle the respirators up to the last detail. To review a thousand protocols, to go over a thousand things, educating myself on a daily basis. I had to leave the ICU and look for information, because since I was isolated, I couldn’t disconnect. 24 h a day thinking about <<what can I do to make this better?>>“. (P. 07F32).
Peer support and teamwork	Emotional support	“During the work shift we would get together, we would go out to the terrace, some of us would stay on duty, when you couldn’t take it anymore, when you had to cry, we would go out to the terrace, with another colleague, always accompanied and we would support each other. Emotional support among colleagues was essential, in the team, among nurses and assistants we supported each other”. (P. 06F52).
	Trust	“That was proper teamwork, I trust you and you trust me. We got along well because we knew how to work as a team.” (P. 04F46).
	Nursing team (nurse and auxiliary nurse care technician)	“As time went by, we realized that we had to change the way we worked and focus on the nurse-assistant team, all working hand in hand. And the truth is that we did it very well, I am very proud of that whole period.” (P. 08M32).
	Support outside working hours	“Above all, with two colleagues we spent the evenings or mornings when we weren’t on shift talking on the phone a lot, this was vital to feel supported and understood… since there were many things you couldn’t tell your family.” (P. 09F52).
Moments of catharsis and disconnection through humor, music, or dance	The media’s distorted image	“When the media talked about the health care workers it was all partying, dancing, laughter, applause… It’ s true that we enjoyed those moments because I myself have danced with my colleagues in the ICU, we played the radio and well, those were moments of disconnection, but what we were experiencing was so intense that it seems that this has not been seen or perhaps they have tried to infantilize people because we were really seeing the worst, what was really happening… however, they come out dancing on television. And it was like… this is not our reality. Those were very specific moments of disconnection, because we need them because it’ s true that people who work in certain special services make certain jokes to try to disconnect, but that’s not what happens during the whole shift. And what we were going through was very strong, then you saw those images and you said <<What kind of image are the public going to have of us?>>, the media is treating the disease as something brutal, people are seeing the morgue opening, the military trucks coming out, the armies disinfecting, and suddenly, we were dancing, I think we were portrayed quite badly.” (P. 07F32).
Focusing on care and everyday work	Focusing on care	“The feeling that another patient is coming, and I have no idea what we are going to do, what we are doing, that this is going to blow up in our face… So, at three or four o’clock in the morning there came a time when we had to focus and say, let’s plan what we are going to do, let’s call the management and start referring patients because this is going to blow up… I think we did well.” (P. 04F46).
Accepting the pandemic as a work situation	Professional duty and commitment	“I haven’t thought about running away. That never crossed my mind. I thought that we were in the situation we were in and that we had to move forward and keep fighting. If I had to catch it, that’s it.” (P. 03F38).
Forgetting bad situations	Dissociative amnesia	“The mind tries not to remember the negative and unpleasant things. I try to evade because I see that at the beginning, I was very tough psychologically, I was very strong but now that all this has happened you fall apart. So, you try not to touch the subject.” (P. 06F52).
Valuing positive reinforcement from patients, their families, and society	Positive reinforcement	“It definitely reinforces your profession to know that patients thank you, that you’ve been there hanging in, that there are people who have left, and you’ve been hanging in there all along.” (P. 04F46). “The gratitude of the people who have been hospitalized and who have recovered in the end. When you made a video call, both the families and the patients were eternally grateful. They didn’t know who you were because they could see your little eyes, but in that sense, it has been gratifying.” (P. 10F30).
	Support for demanding workplace improvements	“The support we were receiving from society through the applause I personally felt comforted and fulfilled, but always hoping that this would become a real support to improve working conditions, contracts… and to improve the visibility and the concept that society has of health care and nurses. Because I have no use for applause in those moments when we have a noose around our necks and all eyes were on us to move forward. If right now it doesn’t turn into salary improvements, labor improvements…” (P. 01M24).
Humanizing the situation	Video calls to family members	“When they started making video calls, I volunteered and made at least four video calls a day. Family members had to request them, and you went with a Tablet, or a cell phone provided by the Health Service. I dedicated many hours to this, instead of leaving at three o’clock, I would leave at half past four, or half past six… whatever time it was.” (P. 06F52).
	The family’s farewell to patients at the end of life	“In cases of terminally ill patients, we provided the relatives with PPE, which we didn’t have a lot of. We tried to humanize the situation a bit. We also made one call a day or every other day with the relatives.” (P. 05F30).

## Data Availability

The data presented in this study are available on request from the corresponding author.

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
