# Peer review of "Nurses’ Perceptions of Ethical Conflicts When Caring for Patients with COVID-19"

_ijerph, 2023, doi:10.3390/ijerph20064763_

Round 1
Reviewer 1 Report
I read the manuscript with great interest. In order to improve the paper, please review the following points.
There are scattered places in the text where periods are missing, please check and correct them carefully. There are unnecessary spaces in sentences. There are also some areas where the text does not have the required spaces. Additionally, there are some areas where there is red text and a history of changes. Please carefully check your use of abbreviations such as PPE, and FFP, etc.
Author Contributions, Funding, Institutional Review Board Statement, Informed Consent Statement, Data Availability Statement, Acknowledgments, and Conflicts of Interest are not listed.
Please check the submission rules carefully.
Line 68-70
The following text is incomplete.
A theoretical sample of nurses who cared for people admitted with COVID-19 during the first and second waves of the pandemic in public hospitals of the regions of XXXXX. and XXXXX was used.
Line 74-76
What is the difference between Socio-demographic characteristics and Main characteristics (Table 1)?
Line 93
Is the expression “Statical analysis” correct?
Line 94-99
Did you read and analyze the original work of Giorgi, A.?
I do not think that the reference in #21 describes the research methodology of Giorgi, A.
I think a careful explanation of the analytical method based on Giorgi, A. original work is necessary.
The results of his analysis depend on the understanding of his analytical method.
Line 131
Table 3.
What does a statement like (P.07M32) at the end of the Verbatims text mean? Does it need to be stated in the analytical method?
Line 410
The following text is incomplete.
This work was sponsored by the University of XXXXXXXX ( GRANT XXXXX)
Author Response
RESPONSE TO REVIEWER #1
Prof. Dr. Paul B. Tchounwou
Editor-in-Chief
International Journal of Environmental Research and Public Health
February 27, 2023
Dear Editor-in-Chief
Enclosed you will find a revision of our manuscript which we now entitle “Nurses' perceptions of ethical conflicts when caring for patients with COVID-19” (Manuscript ID: ijerph-2251548).
We would like to thank you for allowing us to revise and improve our manuscript. We also thank the reviewers for their thoughtful and constructive comments. We have considered all the suggestions and have incorporated them into the revised and updated manuscript.
The article has been revised and modified to improve the understanding of the study. Changes to the original manuscript have been marked in red, and we believe our manuscript is stronger because of these modifications. We hope this now helps to improve the overall readability and quality of the manuscript.
Below is a detailed point-by-point response to the reviewers' comments.
Reviewer: 1
Comments to the Author:
I read the manuscript with great interest. In order to improve the paper, please review the following points.
There are scattered places in the text where periods are missing, please check and correct them carefully. There are unnecessary spaces in sentences. There are also some areas where the text does not have the required spaces. Additionally, there are some areas where there is red text and a history of changes. Please carefully check your use of abbreviations such as PPE, and FFP, etc.
Authors’ response
Many thanks for your comments. We have reviewed the entire document and corrected all mistakes.
Author Contributions, Funding, Institutional Review Board Statement, Informed Consent Statement, Data Availability Statement, Acknowledgments, and Conflicts of Interest are not listed. Please check the submission rules carefully.
Authors’ response
Many thanks. We have added these sections at the end of the manuscript.
Line 68-70. The following text is incomplete. A theoretical sample of nurses who cared for people admitted with COVID-19 during the first and second waves of the pandemic in public hospitals of the regions of XXXXX. and XXXXX was used.
Authors’ response
Many thanks. “XXXXX” represent the name of the provinces, information that is blinded for the review and that will be provided in the final version of the manuscript. We've changed “XXXXX” to "blinded for review".
A theoretical sample of nurses who cared for people admitted with COVID-19 during the first and second waves of the pandemic in public hospitals of the regions of blinded for review and blinded for review was used.
Line 74-76. What is the difference between Socio-demographic characteristics and Main characteristics (Table 1)?
Authors’ response
Many thanks for your comment. Socio-demographic characteristics are included in the main characteristics of participants. We have improved the wording of this section to clarify it in the text and in the table.
The socio-demographic characteristics, and characteristics related to educational level and employment of the participants are shown in Table 1.
Line 93. Is the expression “Statical analysis” correct?
Authors’ response
Many thanks for your comment. We have changed this to “Data analysis”
Line 94-99
Did you read and analyze the original work of Giorgi, A.? I do not think that the reference in #21 describes the research methodology of Giorgi, A. I think a careful explanation of the analytical method based on Giorgi, A. original work is necessary. The results of his analysis depend on the understanding of his analytical method.
Authors’ response
Many thanks for your comments. We apologize that there was a mistake, the correct reference is #17. In this research, Giorgi's original work has been followed (Giorgi, A. (2009). The descriptive phenomenological method in psychology: A modified Husserlian approach. Duquesne University Press), following the steps of Giorgi's phenomenological analysis, as described in the data analysis section.
Line 131. Table 3.What does a statement like (P.07M32) at the end of the Verbatims text mean? Does it need to be stated in the analytical method?
Authors’ response
Many thanks for your comment that allows us to clarify this aspect. Tables 3 and 4 show the categories, codes, and a selection of the main verbalizations. This information is the code associated with each interview that identifies the interview number, the gender (F: female and M: male) and the age of the participant. We have explained this better in the text.
Tables 3 and 4 show the categories, codes, and a selection of the main verbalizations. The verbalizations are accompanied by the code of each interview that identifies the interview number, the gender (F: female and M: male) and the age of the participant.
Line 410. The following text is incomplete. This work was sponsored by the University of XXXXXXXX ( GRANT XXXXX)
Authors’ response
Many thanks. “XXXXXXX” represents the name of the university, this information is blinded for the review and will be provided in the final version of the manuscript. We've changed “XXXXX” to "blinded for review". Moreover, we have added the grant’s data.
Funding: This work was sponsored by the University of blinded for review (GRANT 2021-GRIN-31074).

Reviewer 2 Report
Nurses' perceptions of ethical conflicts of caring patients with COVID-19
It provides important insights into the ethical challenges and dilemmas faced by nurses during the COVID-19 pandemic, as well as the coping strategies they used to deal with them.
Abstract:
Clarify the study design and analysis method. In the current version, it's unclear what type of study was conducted or how the data was analyzed.
The abstract ends somewhat abruptly without clearly stating the consequences of the study. Consider adding a sentence that emphasizes the importance of addressing ethical conflicts and providing psychological support for nurses in the context of the COVID-19 pandemic.
Overall, the abstract provides a solid overview of the study, but with a few improvements, it could be even more effective at communicating the key findings and implications.
The introduction provides a good overview of the ethical challenges and dilemmas that nurses have faced while caring for COVID-19 patients during the pandemic. However, It would be helpful to provide a more detailed background on the ethical issues that nurses have faced during the COVID-19 pandemic, perhaps by referencing relevant literature or statistics.
The Methods section could be to provide more detail on the process of participant recruitment and selection. For example, how were the eligible nurses identified and approached? Were they recruited through their managers or other means? Additionally, it could be helpful to include more information on the sample size and how it was determined, as well as any efforts made to ensure the diversity of participants in terms of demographics and hospital setting. Finally, it may be beneficial to provide more information on how the interviews were conducted, such as the number of interviews per participant and the duration of each interview.
Additionally, the criteria for determining data saturation could be described in more detail, such as how many interviews were conducted overall and how it was determined that additional participants were no longer providing new information.
In terms of the data collection process, it would be helpful to provide more information about how the script of topics for the interviews was developed and refined, such as whether it was based on a literature review or feedback from initial interviews. Finally, some additional details could be given about the training and qualifications of the main researcher who conducted the interviews, including any steps taken to minimize potential biases or preconceptions.
Design and participants: The description of the Giorgi's phenomenological approach used in the study could be expanded to provide more detail about the specific steps involved in the analysis process.
Data collection: The timeframe for data collection (May to December 2020) could be highlighted more clearly to emphasize that the study was conducted during the first two waves of the COVID-19 pandemic.
Statistical analysis: The methods section does not include information about the specific statistical tests or analyses used in the study. It may be useful to provide more detail about how the data were analyzed and if any specific software was used.
Trustworthiness: While the section provides information about strategies used to ensure study trustworthiness, it could be expanded to provide more detail about how these strategies were implemented and what specific steps were taken to ensure the reliability and validity of the study findings.
Clarity: Some parts of the text could benefit from clearer wording or organization. For example, there are a few instances where sentences are quite long and complex, which can make them harder to follow. Breaking up longer sentences or rephrasing them to be more concise could help make the ideas easier to understand.
Structure: While the text is generally organized into sections and subsections, it could benefit from more explicit signposting of how the different themes and sub-themes relate to each other. Providing more explicit transitions between sections or using headings to more clearly signal the main topics could help follow the text's structure more easily.
The discussion provides valuable insights into the ethical and moral concerns and conflicts experienced by nurses during the COVID-19 pandemic. However, the following improvements could be made to enhance its clarity and coherence:
Provide more context: While the discussion provides a detailed account of the experiences of nurses during the pandemic, it would benefit from more contextual information. For instance, the discussion does not specify the location or setting of the study or the sample size, which could provide important context.
Clarify the themes: The discussion outlines two main themes - the ethical conflicts experienced by nurses and the coping strategies used by nurses. However, it would benefit from a more detailed description of these themes.
Provide more detail on coping strategies: The discussion mentions several coping strategies used by nurses, such as support and teamwork, catharsis and disengagement, and focusing on care and everyday work. However, it would benefit from a more detailed description of these strategies, including specific examples of how they were employed by the nurses.
Improve the flow and organization: The discussion covers a wide range of topics related to the ethical and moral concerns and conflicts experienced by nurses during the pandemic. However, the flow and organization could be improved to enhance the text and improve understanding. For example, the discussion could be structured around the two main themes, with more detailed subheadings.
The conclusion can be improved by adding a more definitive call-to-action and by summarizing the main findings of the study in a more concise manner.
Author Response
RESPONSE TO REVIEWER #2
Prof. Dr. Paul B. Tchounwou
Editor-in-Chief
International Journal of Environmental Research and Public Health
February 27, 2023
Dear Editor-in-Chief
Enclosed you will find a revision of our manuscript which we now entitle “Nurses' perceptions of ethical conflicts when caring for patients with COVID-19” (Manuscript ID: ijerph-2251548).
We would like to thank you for allowing us to revise and improve our manuscript. We also thank the reviewers for their thoughtful and constructive comments. We have considered all the suggestions and have incorporated them into the revised and updated manuscript.
The article has been revised and modified to improve the understanding of the study. Changes to the original manuscript have been marked in red, and we believe our manuscript is stronger because of these modifications. We hope this now helps to improve the overall readability and quality of the manuscript.
Below is a detailed point-by-point response to the reviewers' comments.
Reviewer: 2
Comments to the Author:
It provides important insights into the ethical challenges and dilemmas faced by nurses during the COVID-19 pandemic, as well as the coping strategies they used to deal with them.
Authors’ response
Many thanks for this positive comment. We appreciate the acknowledgment of our work.
Abstract:
Clarify the study design and analysis method. In the current version, it's unclear what type of study was conducted or how the data was analyzed.
The abstract ends somewhat abruptly without clearly stating the consequences of the study. Consider adding a sentence that emphasizes the importance of addressing ethical conflicts and providing psychological support for nurses in the context of the COVID-19 pandemic.
Overall, the abstract provides a solid overview of the study, but with a few improvements, it could be even more effective at communicating the key findings and implications.
Authors’ response
Thank you very much for your comment. Following your recommendations, we have added new sentences in abstract in order to clarify these aspects. Please see the changes made in the abstract.
The introduction provides a good overview of the ethical challenges and dilemmas that nurses have faced while caring for COVID-19 patients during the pandemic. However, It would be helpful to provide a more detailed background on the ethical issues that nurses have faced during the COVID-19 pandemic, perhaps by referencing relevant literature or statistics.
Authors’ response
Thank you very much for your comment. Following your recommendations, we have added new sentences to the introduction section with further information.
The Methods section could be to provide more detail on the process of participant recruitment and selection. For example, how were the eligible nurses identified and approached? Were they recruited through their managers or other means? Additionally, it could be helpful to include more information on the sample size and how it was determined, as well as any efforts made to ensure the diversity of participants in terms of demographics and hospital setting. Finally, it may be beneficial to provide more information on how the interviews were conducted, such as the number of interviews per participant and the duration of each interview.
Authors’ response
Thank you very much for your comment. Following your recommendations, we have added new sentences to the methodology section with further information. In relation to the size of the sample, as this was a qualitative investigation, the sample was determined by the criterion of data saturation as specified in the methodology, at which point continuing to expand the sample stopped providing new analytical concepts. This process is guided by the constant comparison method. By carrying out a simultaneous data collection and analysis process, it is possible to know when the data saturation point is reached. See modified text below:
Data collection and data sources
Data collection and analysis was carried out simultaneously following the constant comparison method. Data collection continued until data saturation, at which point additional input (new codes, categories or subcategories) from new participants no longer provides new information (21).
In terms of the data collection process, it would be helpful to provide more information about how the script of topics for the interviews was developed and refined, such as whether it was based on a literature review or feedback from initial interviews. Finally, some additional details could be given about the training and qualifications of the main researcher who conducted the interviews, including any steps taken to minimize potential biases or preconceptions.
Authors’ response
Thank you very much for your comment. Following your recommendations, we have added new sentences to the methodology section to provide more information. The interviewer was an expert in qualitative research and had extensive experience in the semi-structured interview technique. He has a background in Public Health and no clinical experience in the field of study. See modified text below:
The interviews were conducted in a comfortable, private location agreed with the participant by the main researcher (blinded for review), who had a script of topics that could appear openly during the interview (Table 2) (20). The interview scrip of topic was refined throughout the study, following the constant comparison method, with the information that emerged in each interview. The interviewer was an expert in qualitative research and had extensive experience in the semi-structured interview technique. He has a background in Public Health and no clinical experience in the field’s study. The position of the interviewer is a strength of this study, which prevented the interviewer from knowing too much about the phenomenon under study. Moreover, none of the authors knew the participants. Most of the interviews were conducted face-to-face, however, due to the evolution of the pandemic, four participants preferred to conduct the interview by videoconference.
Design and participants: The description of the Giorgi's phenomenological approach used in the study could be expanded to provide more detail about the specific steps involved in the analysis process.
Authors’ response
Thank you very much for your comment. The Giorgi’s phenomenological approach is described in the following paragraph in methodology section. Moreover, we have added the original work of Giorgi, so that any reader can find more information about Giorgi's analysis process. Due to limited space in the publication, we have summarized this process, as shown below:
Once transcribed and anonymized, the interviews were analyzed following Gior-gi's phenomenological method: 1) collecting and describing phenomenological data, 2) reading the entire description, 3) breaking descriptions into meaning units, 4) transforming meaning units, 5) identifying the essential structure of the phenomenon and 6) integrating features into the essential structure of the phenomenon. Theoretical reflective writing was constant during the analysis process (17).
The data analysis was carried out independently by two researchers, who subse-quently reached a consensus on the results; in the event of disagreement, a third re-searcher was used. The research team examines the codes and themes to formulate the categories and final themes. Atlas-ti 8.0 software was used to assist in this phase.
Data collection: The timeframe for data collection (May to December 2020) could be highlighted more clearly to emphasize that the study was conducted during the first two waves of the COVID-19 pandemic.
Authors’ response
Thank you very much for your comment. Following your recommendations, we have added new sentences to the methodology section to provide more information, see changes below:
Design and participants
A qualitative study designed and analyzed using Giorgi's descriptive phenomenological approach, which discover the meaning of a phenomenon through the identification of fundamental themes (17). This approach was chosen to describe the experiences of nurses caring for people with COVID-19 during the first two waves of the COVID-19 pandemic and the ethical dilemmas from the nurses’ experience through a phenomenological analysis of (and in) their own words (18).
…
Semi-structured interviews conducted between May and December 2020 were used for data collection to include nurses’ experiences of the first two waves of the COVID-19 pandemic.
Statistical analysis: The methods section does not include information about the specific statistical tests or analyses used in the study. It may be useful to provide more detail about how the data were analyzed and if any specific software was used.
Authors’ response
Thank you very much for your comment. This is a qualitative study, therefore there is no statistical analysis, but qualitative analysis of the data. In this case the methods of Giorgi's phenomenological analysis were followed. The description of this analysis method is available in the methodology section. Moreover, we used Atlas-ti 8.0 software to assist in this phase. Please, see the methodology section.
Trustworthiness: While the section provides information about strategies used to ensure study trustworthiness, it could be expanded to provide more detail about how these strategies were implemented and what specific steps were taken to ensure the reliability and validity of the study findings.
Authors’ response
Thank you very much for your comment which enables us to expand on this section. New paragraphs have been added to clarify these aspects. See new text below:
The criteria of credibility, transferability, dependability, and confirmability were used to establish study trustworthiness (22). Various strategies have been developed to ensure credibility, such as: reflexivity (all researchers had with a constant reflective attitude attending to methodological crossroads decision making and interpretative issues), research triangulation (comparison and discussion between two different researchers during analytical process. In case of discrepancy, a third investigator was called upon to resolve the discrepancies) and return to interviewees (during the last interviews, as themes were becoming saturated, new questions were introduced in interviews to confirm the intersubjective experience interpretation. In addition, the transcripts of the interviews were returned to the participants to check their agreement). In relation with confirmability, along this manuscript, we have tried to be very transparent with our research process for others researcher to know what stages and how we have cope with different tasks and difficulties. Finally, in terms of helping the audience to assess transferability of results, we have meticulously described the participant characteristics that condition their experience as well as the environment were the research was conducted.
Clarity: Some parts of the text could benefit from clearer wording or organization. For example, there are a few instances where sentences are quite long and complex, which can make them harder to follow. Breaking up longer sentences or rephrasing them to be more concise could help make the ideas easier to understand.
Authors’ response
Thank you very much for your comment. The entire text has been reviewed by a native English speaker specialized in scientific writing. Changes have been made throughout the text to improve English language and grammar. See proofreading certificate attached.
Structure: While the text is generally organized into sections and subsections, it could benefit from more explicit signposting of how the different themes and sub-themes relate to each other. Providing more explicit transitions between sections or using headings to more clearly signal the main topics could help follow the text's structure more easily.
Authors’ response
Thank you very much for your comment. Tables 3 and 4 show the relationship between the different categories, subcategories and codes that emerged in each of the themes, accompanied by the most representative verbalizations of the participants. Space limitations prevent us from explaining this in more detail in the text. We have included new phrases in the results to improve understanding of the relationship between topics, categories, and codes. Please see the results section, and text below:
Tables 3 and 4 show the categories, codes, and a selection of the main verbalizations for each of the emerging themes. The verbalizations are accompanied by the code of each interview that identifies the interview number, the gender (F: female and M: male) and the age of the participant.
1.Theme1. Ethical conflicts during the pandemic
…
2.Theme 2. Coping styles for dealing with ethical conflicts
The discussion provides valuable insights into the ethical and moral concerns and conflicts experienced by nurses during the COVID-19 pandemic. However, the following improvements could be made to enhance its clarity and coherence: Provide more context: While the discussion provides a detailed account of the experiences of nurses during the pandemic, it would benefit from more contextual information. For instance, the discussion does not specify the location or setting of the study or the sample size, which could provide important context.
Authors’ response
Many thanks for this positive comment. We appreciate the acknowledgment of our work. Following your recommendations, we have added more information about this in the discussion section.
Clarify the themes: The discussion outlines two main themes - the ethical conflicts experienced by nurses and the coping strategies used by nurses. However, it would benefit from a more detailed description of these themes.
Authors’ response
Thank you very much for your comment. We have provided more information on this in the discussion section.
Provide more detail on coping strategies: The discussion mentions several coping strategies used by nurses, such as support and teamwork, catharsis and disengagement, and focusing on care and everyday work. However, it would benefit from a more detailed description of these strategies, including specific examples of how they were employed by the nurses.
Authors’ response
Thank you very much for your comment.
Improve the flow and organization: The discussion covers a wide range of topics related to the ethical and moral concerns and conflicts experienced by nurses during the pandemic. However, the flow and organization could be improved to enhance the text and improve understanding. For example, the discussion could be structured around the two main themes, with more detailed subheadings.
Authors’ response
Many thanks for this positive comment. We appreciate the acknowledgment of our work. Following your recommendations, we have added structured the discussion around the two main themes.
The conclusion can be improved by adding a more definitive call-to-action and by summarizing the main findings of the study in a more concise manner.
Authors’ response
Thank you very much for your comment. We have added new sentences to improve the conclusion.
CONCLUSION
Nurses have experienced ethical conflicts on a personal and professional level and have striven to optimize and deliver safe and ethical care to patients with COVID-19 even when their lives were at risk. The main coping strategies used by nurses to deal with these conflicts have been active and autonomous learning, peer support and teamwork, moments of emotional catharsis, focusing on care, accepting the pandemic as just another work situation, forgetting bad situations experienced, valuing positive reinforcement from patients, relatives, and society, and humanizing the situation. Strong professional duty and commitment, teamwork, humanization of care, and continuing education have helped nurses cope with such conflicts.
Ethical concerns and conflicts that arose along with work overload and other fac-tors may have contributed to increased psychological distress among frontline nurses.
Future studies should analyze ethical conflicts arising in the context of primary care and social and health care. In addition, it would be useful to know which interventions would lead to improvements in the coping strategies used by nurses during ethical conflicts.
Implications for Nursing Management
Ethical concerns and conflicts that arose along with work overload and other fac-tors may have contributed to the increase in psychological distress among frontline nurses. It is necessary to address ethical conflicts and provide psychological and emotional support programs for those nurses who have experienced personal and professional ethical conflicts in trying to optimize and provide safe and ethical care during COVID-19.

Round 2
Reviewer 1 Report
I believe that the manuscript has been improved in line with reviewers comments. Thank you very much.
In Table 1, "Participant characteristics (N=14)" is redundant and unnecessary.
The title of table 1 could be changed to "Table 1. main characteristics of participants (N=14)."
In Table 2, "Interview topic script" is also unnecessary because it is duplicated.
As for Tables 3 and 4, please place the titles above the tables.
On page 14, line 142, there is no period. Please check and correct.
Author Response
RESPONSE TO REVIEWER #1
Prof. Dr. Paul B. Tchounwou
Editor-in-Chief
International Journal of Environmental Research and Public Health
March 02, 2023
Dear Editor-in-Chief
Enclosed you will find a revision of our manuscript which we now entitle “Nurses' perceptions of ethical conflicts when caring for patients with COVID-19” (Manuscript ID: ijerph-2251548).
We would like to thank you for allowing us to revise and improve our manuscript. We also thank the reviewers for their thoughtful and constructive comments. We have considered all the suggestions and have incorporated them into the revised and updated manuscript.
The article has been revised and modified to improve the understanding of the study. Changes to the original manuscript have been marked in red, and we believe our manuscript is stronger because of these modifications. We hope this now helps to improve the overall readability and quality of the manuscript.
Below is a detailed point-by-point response to the reviewers' comments.
Reviewer: 1
Comments to the Author:
I believe that the manuscript has been improved in line with reviewers comments. Thank you very much.
Authors’ response
Thank you very much for your positive consideration of our manuscript.
In Table 1, "Participant characteristics (N=14)" is redundant and unnecessary.
Authors’ response
Thank you very much for your positive consideration of our manuscript.
The title of table 1 could be changed to "Table 1. main characteristics of participants (N=14)."
Authors’ response
Many thanks for your comment. We have changed this to “Table 1. Main characteristics of participants (N=14).”
Table 1. Main characteristics of participants (N=14).
In Table 2, "Interview topic script" is also unnecessary because it is duplicated.
Authors’ response
Thank you very much for your comment. We have deleted that sentence.
As for Tables 3 and 4, please place the titles above the tables.
Authors’ response
Thank you very much for your comment. We have placed the titles of those tables in the correct place. Please see the changes made in red in tables 3 and 4.
On page 14, line 142, there is no period. Please check and correct.
Authors’ response
Thank you very much for your comment. On page 14? Or on page 4? Even so, we do not understand what you mean, if you can clarify it for us, we would appreciate it.

Reviewer 2 Report
Dear Authors
Thank you for accepting the proposed improvements. Best of Luck to you!
Author Response
RESPONSE TO REVIEWER #2
Prof. Dr. Paul B. Tchounwou
Editor-in-Chief
International Journal of Environmental Research and Public Health
March 02, 2023
Dear Editor-in-Chief
Enclosed you will find a revision of our manuscript which we now entitle “Nurses' perceptions of ethical conflicts when caring for patients with COVID-19” (Manuscript ID: ijerph-2251548).
We would like to thank you for allowing us to revise and improve our manuscript. We also thank the reviewers for their thoughtful and constructive comments. We have considered all the suggestions and have incorporated them into the revised and updated manuscript.
The article has been revised and modified to improve the understanding of the study. Changes to the original manuscript have been marked in red, and we believe our manuscript is stronger because of these modifications. We hope this now helps to improve the overall readability and quality of the manuscript.
Below is a detailed point-by-point response to the reviewers' comments.
Reviewer: 2
Dear Authors
Thank you for accepting the proposed improvements. Best of Luck to you!read the manuscript with great interest.
Authors’ response
Thank you very much for your positive consideration of our manuscript.
